# Effect of Glue Spread on Bonding Strength, Delamination, and Wood Failure of Jabon Wood-Based Cross-Laminated Timber Using Cold-Setting Melamine-Based Adhesive

**DOI:** 10.3390/polym15102349

**Published:** 2023-05-17

**Authors:** Yusup Amin, Renaldi Purnomo Adji, Muhammad Adly Rahandi Lubis, Naresworo Nugroho, Effendi Tri Bahtiar, Wahyu Dwianto, Lina Karlinasari

**Affiliations:** 1Research Center for Biomass and Bioproducts, National Research and Innovation Agency, Cibinong 16911, West Java, Indonesia; 2Department of Forest Products, Faculty of Forestry and Environment, IPB University, Bogor 16680, West Java, Indonesia

**Keywords:** bonding strength, citric acid, delamination, jabon wood, melamine-formaldehyde, pMDI

## Abstract

Cross-laminated timber (CLT) has become a popular engineered wood product due to its innovative properties and rapid development, which involves the use of various wood species and adhesives. This study aimed to assess the effect of glue application on the bonding strength, delamination, and wood failure of CLT made from jabon wood and bonded with a cold-setting melamine-based adhesive at three different rates: 250, 280, and 300 g/m^2^. The adhesive was composed of melamine–formaldehyde (MF) by adding 5% citric acid, 3% polymeric 4,4-methylene diphenyl diisocyanate (pMDI), and 10% wheat flour. Adding these ingredients increased the adhesive viscosity and decreased the gelation time. The CLT samples, made using cold pressing in the melamine-based adhesive at a pressure of 1.0 MPa for 2 h, were evaluated as per the standard EN 16531:2021. The results revealed that a higher glue spread resulted in a greater bonding strength, lower delamination, and a higher wood failure. The glue spread was shown to have a more significant influence on wood failure compared with delamination and the bonding strength. The application of 300 g/m^2^ glue spread (MF-1) on the jabon CLT led to a product that met the standard requirements. The use of modified MF in cold-setting adhesive produced a potential product that could be a feasible option for future CLT production in terms of its lower heat energy consumption.

## 1. Introduction

Engineered wood products have gained popularity worldwide due to the increasing demand for environmentally friendly building construction and the growth of global development [1,2,3,4]. Cross-laminated timber (CLT) is an example of this type of product, which has emerged as a viable building material for structural applications. CLT is produced by laminating graded sawn lumber, with each layer placed orthogonally and bonded with structural adhesives [5,6]. As a building material for structural applications [7,8], it has great potential for multistory constructions and high-rise buildings [9]. The mechanical and physical characteristics of cross-laminated timber (CLT) are better than those of other types of manufactured wood products, such as glued laminated timber, oriented strand board, and laminated veneer lumber [9,10].

Recently, there has been increased interest in the global production of CLT, particularly outside of Europe [4,9,10,11]. Even though Japan and China have started CLT production, in countries such as Malaysia and Indonesia, it is still in the early stages of development [10,12]. Hence, extensive research and development is required to produce CLT from local wood species. The selection of appropriate raw materials is a crucial prerequisite for ensuring high-quality CLT production. It is necessary to take into account factors such as the species availability, strength properties, bonding characteristics, and durability when manufacturing CLT using local wood species [12]. The CLT production process can differ between countries owing to variations in tree species and available adhesives. Therefore, the parameters involved in the production from local tree species, including the type of adhesive, glue consumption, and pressure, should be explored [7].

CLT production uses layers from wood species to enhance efficiency and improve quality. In the manufacturing process, adhesive selection is a crucial factor that requires careful consideration. The types of adhesives ordinarily used to fabricate CLT include phenol–resorcinol formaldehyde, melamine formaldehyde (MF), emulsion polymer isocyanate, and polyurethane [13]. Melamine–urea formaldehyde and polyurethane adhesives are used in European production [2,11], while MF, phenol–resorcinol formaldehyde, and isocyanate adhesives are commonly used in Japan [2]. MF adhesives are extensively used in the production of semiexterior- and exterior-grade wood-based panels [14]. The adhesive is widely selected by glulam or CLT producers because of its relatively low cost, transparent bonding lines, excellent durability, heat resistance, and resistance to water and moisture [15,16]. However, the adhesive bonds must be capable of transferring loads and maintaining their strength and durability over the service life of the structure [17]. As a thermosetting adhesive, MF is generally applied with the use of hot presses, which involves additional costs due to heat energy consumption. Merline et al. [18] showed that the curing temperature of MF adhesive is 140–160 °C. Furthermore, heat treatment during wood composite production can lead to the emission of formaldehyde, which is produced from adhesives or wood as a raw material [19,20].

Innovation is needed to modify MF-based adhesives to allow cold-pressing applications with reduced formaldehyde emissions without decreasing the mechanical strength of the CLT products. In this research, jabon (*Anthocephalus cadamba* Roxb. Miq.), a lightweight tropical wood species found in Indonesia, and a modified MF-based adhesive were used to manufacture CLT through a cold-pressing application process. Jabon wood is a promising candidate for engineered wood products due to its rapid growth and potential availability within local communities [21,22]. In order to improve the production of cross-laminated timber (CLT), a new cold-setting adhesive formula was developed by incorporating polymeric 4,4-methylene diphenyl diisocyanate (pMDI) into the melamine formaldehyde (MF) adhesive. The pMDI resin has demonstrated its effectiveness in producing wood panel products, such as particle boards and medium-density fiberboard [23,24]. The optimal amount of adhesive used in CLT manufacturing should also be considered a critical variable that determines the mechanical performance and production cost [25]. This study aimed to evaluate the delamination, bonding strength, and wood failure of CLT products made from jabon wood using the cold-setting adhesive based on MF and pMDI with glue spreads of 250 g/m^2^, 280 g/m^2^, and 300 g/m^2^.

## 2. Materials and Methods

### 2.1. Materials

The commercial melamine adhesive (MA-204) was supplied by PT Pamolite Adhesive Industry, Probolinggo, East Java, Indonesia. Wheat flour, technical-grade citric acid powder, and polymeric 4,4-methylene diphenyl diisocyanate (pMDI) were obtained from the commercial market and provided by the Integrated Laboratory of Bioproducts (iLab), National Research and Innovation Agency (BRIN), Cibinong, Indonesia. The raw material for producing CLT was obtained from jabon wood, which was obtained from a community forest located in the Bogor area of West Java. The wood had a density of 0.44 g/cm^3^ and a moisture content of 12 ± 2%. To create the CLT, lumber samples with dimensions of 330 mm × 110 mm × 20 mm and 110 mm × 110 mm × 20 mm for the length, width, and thickness were prepared.

### 2.2. Preparation of Melamine-Based Adhesive

Two types of melamine-based adhesive were prepared, i.e., a commercial adhesive as the control (MF-0) and the modified (MF-1) formulation. To modify the commercial melamine adhesive (MF-0), several components were added (MF-1), including wheat flour as a filler, citric acid (20 wt%) as a catalyst, and pMDI (96.2 wt%) as a cross-linker. Citric acid and pMDI were added based on the solids content of the control adhesive: 5% and 3%, respectively. Wheat flour comprised as much as 10% of the total mixture. The resulting mixture was then manually stirred for 1–2 min at 27 ± 2 °C, yielding a cold-setting melamine-based adhesive suitable for use with CLT.

### 2.3. Characterization of Adhesive

A digital pH meter from Laqua, Horiba, Kyoto, Japan was used to measure the pH levels of MF-0 and MF-1. About 50 g of the adhesive sample was placed in a 100 mL glass beaker, and the pH value was recorded after 3–5 min, following which the measurement was repeated three times [26].

To determine the nonvolatile solids content of MF-0 and MF-1, the weight of each adhesive was measured before and after drying. Approximately 2 g of each adhesive was placed in a disposable aluminum foil container and dried for three hours at a temperature of 105 °C. Following the drying process, the weight of the oven-dried adhesive was divided by its initial weight [27], and the solids content was measured in triplicate.

To measure the viscosity of each adhesive, around 20 mL of each adhesive was poured into a measuring cup (C-CC27, Anton Paar, Graz, Austria) and placed onto a rotational rheometer (RheolabQC, Anton Paar, Graz, Austria). The average viscosity of the adhesives was measured using a spindle-type concentric cylinder (CC), number 27, at a constant shear rate of 150/s and a temperature of 25 °C. The viscosity value was obtained using Rheo-Compass software [27]. Furthermore, using Rheo-Compass software (Version 1.30, Anton Paar, Graz, Austria), the dynamic viscosity and cohesion strength of the melamine-based adhesives were measured at a constant shear rate of 150/s at 25 °C for a duration of 200 min.

To determine the gelation time of the melamine-based adhesives, a gel time meter (Techne GT-6, Coleparmer, Vernon Hills, IL, USA) was utilized. A sample of the adhesive was placed in a test tube, and a needle was submerged into it. The time taken for the adhesive to solidify at a temperature of 100 ± 3 °C was measured and recorded in triplicate. The gelation times of MF-0 and MF-1 were also verified using the same method. In this case, approximately 50 g of adhesive was placed in the test tube with the meter needle submerged in the sample, and the time required for the adhesive to solidify at a temperature of 100 ± 3 °C was observed and recorded in triplicate.

The functional groups present in the melamine-based adhesives were analyzed using Fourier-transform infrared spectroscopy (FTIR) with the SpectrumTwo instrument from PerkinElmer, located in Hopkinton, MA, USA. The MF-0 and MF-1 samples were scanned at wavenumbers of between 400 and 4000 cm^−1^ with a resolution of 4 cm^−1^ and 16 scans per sample. The adhesive spectrum was normalized using min–max normalization with Ver. 10.5.3 software from Perkin Elmer Inc., Hopkinton, MA, USA [27].

To investigate the thermo-mechanical properties of the melamine-based adhesives, a dynamic mechanical analysis (DMA 8000, Perkin Elmer Inc.) was conducted. The adhesives were used to bond two Whatman filter papers with a glue spread of 300 g/m^2^ to create a specimen with dimensions of 50 mm × 8 mm × 0.2 mm. Before the DMA analysis, the specimens were pre-cured in an oven at 50 °C for 5 min. The storage modulus (E′), loss modulus (E″), and tan delta of each specimen were determined using a frequency of 1 Hz, a strain level of 0.01%, and a heating rate of 5 °C/min within a scanning range of 20–60 °C [28].

### 2.4. Preparation of CLT

The fabrication of CLT was based on EN 16531:2021 for Timber Structures—CLT Requirements [29]. The three-layer jabon CLT was manufactured using melamine-based adhesives. In this research, longitudinal laminations were constructed from jabon wood measuring 330 mm × 110 mm × 20 mm, with a total of 18 lumbers utilized. Transverse laminations, consisting of 27 lumbers of 110 mm × 110 mm × 20 mm in size were also employed. Control CLT bonded with commercial MF adhesive (MF-0) was prepared at different glue spreads of 250, 280, and 300 g/m^2^ using cold pressing at 1 MPa under clamping for 2–4 min and then subjected to heat treatment in the oven at 100 °C for 2 h (Figure 1). The MF-0 itself could not be used as cold-setting adhesive; therefore, the CLT had to be prepared using hot-pressing (heat treatment). In contrast, CLT-bonded MF-1 was prepared using a cold press at 1 MPa for 2 h, based on preliminary trials and following previous research [12,30]. As with the MF-0 adhesive, different glue spreads were applied for MF-1 bonded CLT, namely 250, 280, and 300 g/m^2^. The preparation of CLT bonded with cold-setting melamine-based adhesives (MF-1) is illustrated in Figure 2.

### 2.5. Evaluation of CLT Properties

To conduct delamination tests in accordance with EN 16531:2021 [29], the CLT samples were cut into 100 mm × 100 mm × 60 mm dimensions. Three specimens were tested for each glue application, and they were submerged in ambient-temperature water within a pressure vessel. A vacuum of 85 kPa was applied and maintained for 30 min before releasing it and applying a pressure of 600–700 kPa, which was held for 2 h. After that, the test pieces were dried for approximately 12–20 h at a temperature of 65 °C. For each specimen, the total delamination and maximum delamination were recorded, and the percentage of delamination was calculated using Equations (1) and (2) [12].
(1)$${\mathrm{Delamination}}_{\mathrm{tot}}=100\;\frac{l_{\mathrm{tot}.\mathrm{delam}}}{l_{\mathrm{tot}.\mathrm{glue}\;\mathrm{line}}}$$
(2)$${\mathrm{Delamination}}_{\max}=100\;\frac{l_{\max.\mathrm{delam}}}{l_{\mathrm{glue}\;\mathrm{line}}}$$
where *l*_tot,delam_ is the total delamination length, *l*_tot,glueline_ is the sum of the perimeters of all glue lines in a delamination specimen, *l*_max,delam_ is the maximum delamination length, and *l*_glueline_ is the perimeter of one glue line in a delamination specimen.

An assessment of the bonding strength of cross-laminated timber (CLT) was conducted by performing a block-shear strength analysis using a previously published method [31]. The CLT was cut into a block-shear sample, as demonstrated in Figure 3, and the block-shear strength test was carried out utilizing a 50 kN universal testing machine (UTM AG-IS 50 kN, Shimadzu, Kyoto, Japan) at a crosshead speed of 2 mm/min. Additionally, in accordance with the EN 16,351 standard [29], the bonding strength was evaluated, and three block-shear samples were tested for each glue spread.

### 2.6. Statistical Analysis

The study compared the average values of delamination, bonding strength, and wood failure of cross-laminated timber (CLT) produced from MF-1 adhesive using an analysis of variance (ANOVA). The Duncan multiple range test at a significance level of α = 0.05 was performed to identify the significant effect of glue spread on the delamination and bonding strength. The statistical analysis was carried out with SPSS 21 software (SPSS Inc., Chicago, IL, USA).

## 3. Results and Discussion

### 3.1. Properties of Cold-Setting Melamine-Based Adhesives

Table 1 shows the characteristics of adhesives made from melamine adhesives. The nonvolatile solids content of MF-0 and MF-1 was about 51 and 48, respectively. This content represents the total solids that contribute to the adhesive coverage on the bonded products. According to a previously published study [32], the nonvolatile solids content of melamine-based adhesives increases with a higher concentration of melamine. The inclusion of citric acid, pMDI, and wheat flour in the melamine resin mix led to a balance in the physical and chemical properties of the adhesive, resulting in a reduction of the solids content of the resin. After the addition of citric acid, pMDI, and wheat flour to MF-0, the solids content of MF-1 decreased. This outcome aligns with a previous study, which reported a decrease in the solids content of melamine-based adhesives when mixed with NH_4_Cl as a hardener [32].

The gelation time in wood adhesive refers to the duration needed for the adhesive molecules to form a gel. This gel formation creates stronger molecular connections between the molecules, resulting in stronger adhesion. As the gel forms, the viscosity of the adhesive increases, and the adhesive becomes less smooth. This point, where the viscosity is exceptionally high and the adhesive forms a gel, is known as the gel point. The gelation time of MF-0 was found to be five times longer than that of MF-1. This difference can be attributed to the absence of a citric acid catalyst or pMDI cross-linker in MF-0, while MF-1 was formulated by adding both citric acid and pMDI to MF-0. The introduction of these catalysts and cross-linkers accelerates the gelation time by promoting the formation of longer molecular chains. This finding is in line with a previous study that reported a reduction in the gelation time of melamine-based adhesives when mixed with NH_4_Cl as a hardener [32]. A study by Lubis et al. [33] reported that incorporating 1–2% pMDI can reduce the gelation time of MF-based adhesives.

The inclusion of 5% citric acid and 3% pMDI into MF-0 resulted in a lower pH value for MF-1 compared to that of MF-0, with pH values of 6.67 and 7.67, respectively. This pH difference can be attributed to the presence of the carboxylic acid group (–COOH) in citric acid and the isocyanate group (–NCO) in pMDI, which can react with free-formaldehyde (HCHO_free_) and the methylol group (–CH_2_OH) to form a cross-linked polymer in melamine-based adhesives.

The average viscosity is a measure of the resistance of a material to gradual deformation by pressure. In the case of liquid adhesives, the average viscosity is related to the thickness of the liquid. The control adhesive, MF-0, had a viscosity of 484.51 mPa∙s, indicating the absence of a catalyst and a cross-linker. Meanwhile, the viscosity of MF-1 was higher than that of MF-0 due to the formation of a bigger molecular chain from the interactions of MF, citric acid, and pMDI. A previous study [33] reported that the viscosity of MF-based adhesive increased by adding 2% pMDI.

The dynamic viscosity and cohesive strength of melamine-based adhesives were analyzed using a rotational rheometer, as depicted in Figure 4a. In line with the result for the average viscosity, the dynamic viscosity of MF-1 was higher than that of MF-0. The dynamic was in the range of 545–587 mPa∙s during 200 min of measurement, while that of MF-0 was around 473–489 mPa∙s. This result shows that the viscosity of MF-1 increased remarkably compared with that of MF-0 in 200 min. Furthermore, the molecules of MF-0 were relatively stable for 200 min. The cohesion strength of an adhesive is related to its viscosity, as a higher viscosity indicates a greater resistance to deformation, which can lead to stronger internal bonds within the adhesive. In this study, the higher viscosity of MF-1 resulted in a higher cohesion strength compared with that of MF-0. MF-1 had a cohesion strength of 82–89 Pa, which was higher than that of MF-0 at about 71–73 Pa (Figure 4b). Therefore, CLT bonded with MF-1 is expected to have better properties than MF-0.

Despite variations in the type of adhesive used, the functional groups present in primary amides, such as –N–H_2_, C–H, C=O, and –N–H, were identified through their respective wavenumbers of 3325, 2960, 1640, and 1565 cm^−1^. The FTIR spectra did not reveal any differences between the melamine-based adhesives (Figure 5). However, following the curing process, the functional groups were modified slightly due to the formation of a three-dimensional network resulting from the reaction between the melamine-based adhesive, catalyst, and cross-linker. A lower intensity of –N–H_2_ at 3325 cm^−1^ was observed, while the C–H, C=O, and N–H functional groups of the primary amide were detected at 2960, 1640, and 1565 cm^−1^ after curing. Theoretically, the cross-linked polymer in melamine-based adhesives was formed by the reaction of –COOH in citric acid and –NCO in pMDI with HCHO_free_ and the –CH_2_OH group [26,27].

DMA was carried out in order to explore the thermo-mechanical characteristics of the melamine-based adhesives, as presented in Figure 6. This method assesses the mechanical responses of viscoelastic substances subjected to oscillation at different temperatures. In addition, a comparison was made between the storage modulus (E′), loss modulus (E″), and tan delta (the ratio of E″ to E′) of the adhesives. The E′ parameter measures the amount of energy stored in the material, which is dependent on the type of polymer, temperature, and oscillation frequency. On the other hand, E″ evaluates the amount of energy dissipated by the specimen as a result of the molecular friction in the viscous flow [28]. The maximum E′, 467.9 GPa, was detected in MF-1 at 38.4 °C, with E′ tending to decrease as a function of an increasing temperature. The initial decline in E′ is attributed to the adhesive becoming softer as the temperature rises. Additionally, E′ increased towards its peak, which could be attributed to the adhesive gelation process in which an extensive molecular network was formed. The trend for E″ was similar to that of E′, and the initial decline was due to adhesive softening. Furthermore, E″ increased after reaching its minimum value, and this finding also reflected the gelation process, as indicated by the tan delta. This reaction led to the formation of a network which, in turn, resulted in efficient energy dissipation. However, the maximum E′ of MF-0 could not be detected, probably due to the absence of hardening and the cross-linking reaction that occurred without the presence of a catalyst and a cross-linker.

### 3.2. Bonding Strength and Delamination of CLT

For CLT manufacturers and consumers, the most important factor is the bond integrity, which involves the bonding strength and delamination [5,34,35,36]. The bond integrity refers to the adhesive strength between two layers and is influenced by the type of adhesive, the wood species used, and the surface treatment [13]. The bonding strength can be determined through block-shear strength testing using a universal testing machine.

A new adhesive formula had to be developed, since commercial MF cannot be directly applied for CLT cold pressing. Our trial study revealed that commercial MF with cold pressing was not successful in assembling CLT. As a result, a control CLT (MF-0) was manufactured using a melamine-based adhesive, which was cold pressed at 1 MPa, clamped, and then heat-treated at 100 °C for 2 h to cure the adhesive, as illustrated in Figure 1. Meanwhile, the properties of CLT bonded with MF-1 were successfully tested using the cold-press method (Figure 2). The result of block-shear testing found that the bonding strength of MF-0 with glue spreads of 300 g/m^2^, 280 g/m^2^, and 250 g/m^2^ were 2.13 MPa, 1.94 MPa, and 1.03 MPa, respectively. Those samples of MF-0 had values about 30% lower than those of the MF-1 samples (Figure 7a). The highest bonding strength of CLT bonded with MF-1 was 3.14 MPa, which was achieved with a glue spread of 300 g/m^2^. Bonding strengths of 2.57 MPa and 1.56 MPa were obtained with glue spreads of 280 g/m^2^ and 250 g/m^2^, respectively. Figure 7b shows the load–displacement curve and block-shear strength of CLT bonded with MF-1. A greater maximum load was observed at a greater displacement as the glue spread increased. A maximum load of approximately 9750 N at a displacement of 8.3 mm was achieved using 300 g/m^2^ of adhesive with MF-1, while 280 g/m^2^ of glue spread led to a maximum load of 5506 N at a maximum displacement of 5.9 mm, and 250 g/m^2^ of glue spread resulted in a maximum load of approximately 3400 N at a displacement of 3.6 mm.

The bonding strength of CLT can be increased by adding 5% citric acid and 3% pMDI, which leads to the formation of a larger molecular chain through the interactions of MF, citric acid, and pMDI. The bonding strength of CLT with MF-1 adhesive is greater than that of CLT with MF-0 adhesive due to the extensive molecular chain of MF-1 and the formation of high cohesion strength. A glue spread of 300 g/m^2^ and a cold pressure of 1 MPa resulted in a better block-shear strength value for jabon CLT bonded with MF-1 (3.14 MPa) than that of cross-laminated bamboo and timber (CLBT) made from flattened bamboo boards and Chinese fir with phenol resorcinol formaldehyde adhesive in a major direction test (2 MPa) [34]. Yang et al. [34] also reported that the block-shear strength of CLBT bonded with one-component polyurethane with a 200 g/m^2^ glue spread and a bonding pressure of 1 MPa was 2.47 MPa in the major direction test. The block-shear strength values of jabon CLT with MF-1 at 280 and 300 g/m^2^ with a pressure of 1 MPa were better than those of CLT made from spruce–pine–fir bonded with polyurethane, emulsion polymer isocyanate, and phenol resorcinol formaldehyde at glue spreads of 200 g/m^2^, 270 g/m^2^, and 250 g/m^2^, respectively [37].

The shear performance of CLT panels is a critical factor in panel failure, as the weakened strength in the transverse layers is perpendicular to the major axis of the panel [8]. In CLT block-shear testing, the leading mode of failure was perpendicular to the grain, resulting in a lower block-shear strength compared with that of glued laminated timber made from the same wood material [25,34]. The research conducted found that the block-shear strength of jabon CLT bonded with both MF-0 and MF-1 did not meet the EN 16531:2021 standard [29], which states that the bonding strength should be at least 6 MPa. However, the bond integrity seemed to have been formed at the MF-1 adhesive applied in a cold setting.

Figure 8a shows the delamination values of jabon CLT bonded with MF-0 and MF-1 adhesives at different glue spreads, while Figure 8b displays the wood failure percentage of the block-shear strength samples after testing. The average delamination of MF-0 for three different glue spreads (250, 280, and 300 g/m^2^) was 26.30–13.07%. In terms of the bonding strength, delamination decreased as the wood failure percentage increased with more glue spread [38,39]. The wood failure percentage of CLT with the MF-0 adhesive had a value of lower than 20%. The delamination and wood failure percentage of MF-0 did not meet the values stipulated in the EN 16531:2021 standard [29]. The block-shear testing samples of CLT bonded with MF-0 adhesive revealed that mostly only glue line damage was found, as shown in Figure 9a. Meanwhile, the delamination and wood failure percentage of MF-1 were better than those of MF-0. The damage after block-shear testing was identified as wood failure (Figure 9b), with the percentage of CLT bonded with MF-1 being nearly 3–5 times higher than that of MF-0, as shown in Figure 8b. Application of 300 g/m^2^ of glue spread (MF-1) resulted in the lowest and highest delamination and wood failure percentage at 5.70% and 98.33%, respectively. The delamination and wood failure percentage of jabon CLT with MF-1 (300 g/m^2^) met the requirements of the EN 16531:2021 standard [29]. The standard specifies that total delamination should be less than 10%, and the wood failure percentage should be at least 90%. Meanwhile, glue spreads of 280 g/m^2^ and 250 g/m^2^ produced CLT with 13.91% and 19.42% delamination and 66.67% and 33.33% wood failure.

The statistical analysis is presented in Table 2. This aimed to evaluate how glue spread affects the bonding strength, delamination, and wood failure percentage of cold-setting melamine-based adhesives (MF-1) applied to jabon wood CLT. The statistical analysis revealed that the wood failure was considerably impacted by the glue spread, with a higher R^2^ of 0.8729, a lower *p*-value of 0.0002, and a higher F-value of 48.0947 compared with the bonding strength and delamination. Conversely, the bonding strength was least affected because of the high standard deviation, with an R^2^ of 0.4728. A multiple range test using Duncan’s post-hoc analysis demonstrated that CLT with an improved bonding strength, delamination, and wood failure could be produced using 300 g/m^2^ of glue spread compared with values of 250 g/m^2^ and 280 g/m^2^.

The specimen exhibited a high level of wood failure and a low delamination percentage, indicating the development of bond integrity, which relies on the integrity of wood adhesive bonds [13]. A study by Lubis et al. [33] found that the properties and reactivity of a modified MF-based adhesive could be enhanced by the addition of citric acid and pMDI. Therefore, our study of a modified MF adhesive with the addition of 5% citric acid and 3% pMDI in a cold-pressing application provides a prospective method in terms of producing high-performance products and supporting eco-friendly programs through the efficient consumption of heat energy in the CLT manufacturing process.

## 4. Conclusions

This research developed CLT from jabon wood made with a modified MF adhesive by adding 5% citric acid and 3% pMDI in a cold-setting application. This addition increased the viscosity and cohesion strength of the melamine-based adhesive and decreased the solids content, gelation time, and pH. CLT products with various glue spreads of 250, 280, and 300 g/m^2^ were investigated to characterize the bonding strength, delamination, and wood failure percentage. The findings indicate that a higher glue spread leads to an increased bonding strength, lower delamination value, and higher wood failure percentage, which are indicators of the high performance of the bondability system. The statistical analysis found that the glue spread has a more significant impact on wood failure than the bonding strength and delamination. Through our study, the MF adhesive mixed with citric acid and pMDI was shown to be a promising cold-setting adhesive that considers heat energy consumption for use in future CLT production.

## Figures and Tables

**Figure 1 polymers-15-02349-f001:**
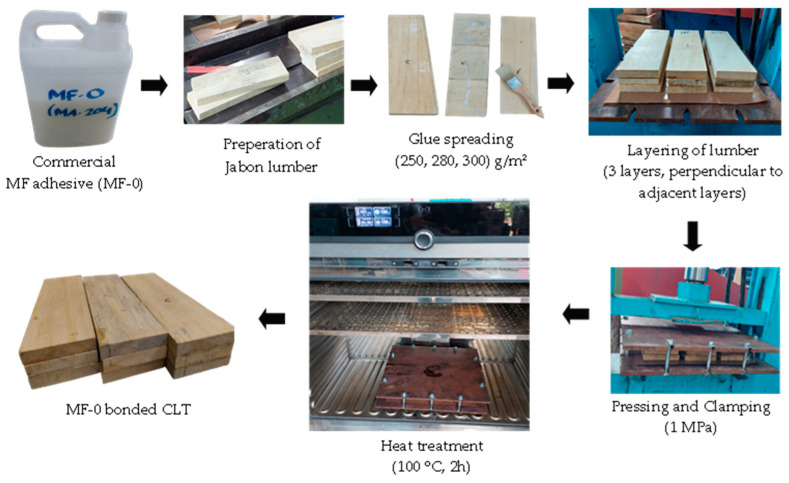
The preparation of CLT bonded with MF-0 using heat treatment.

**Figure 2 polymers-15-02349-f002:**
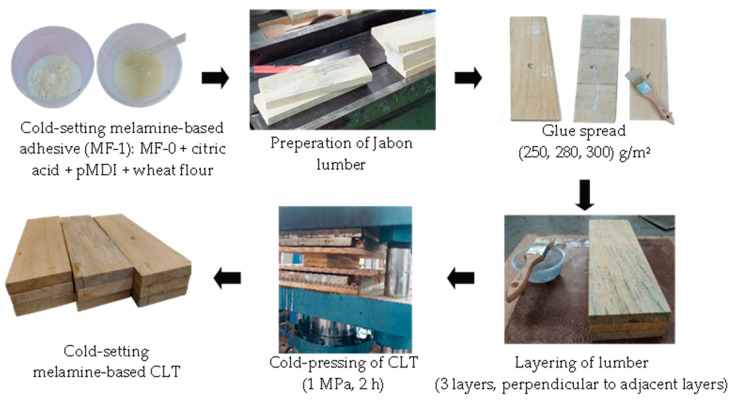
The preparation of CLT bonded with MF-1 using cold pressing.

**Figure 3 polymers-15-02349-f003:**
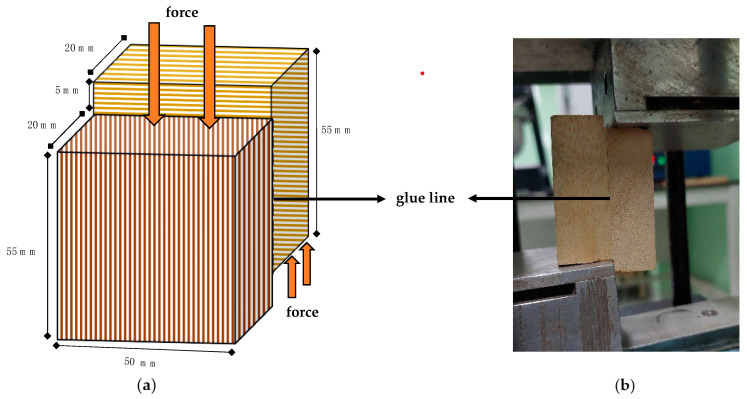
Illustration of the block-shear test of CLT based on the Zhou et al. [31] (**a**), and the block shear testing (**b**); the arrow line points out the glue line.

**Figure 4 polymers-15-02349-f004:**
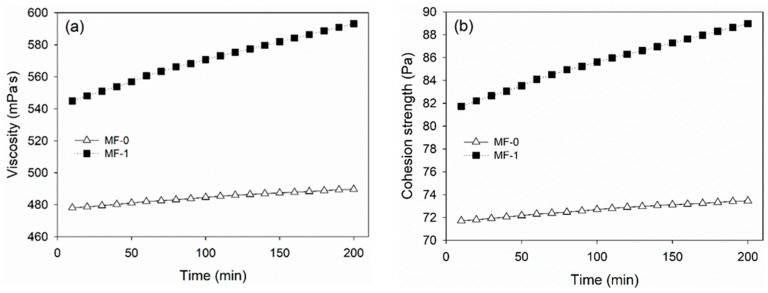
Dynamic viscosity (**a**) and cohesion strength (**b**) of cold-setting melamine-based adhesives.

**Figure 5 polymers-15-02349-f005:**
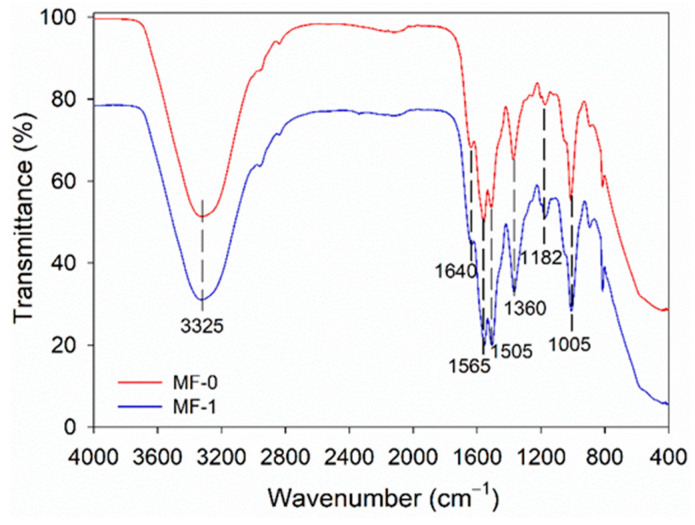
FTIR spectra of the cold-setting melamine-based adhesives.

**Figure 6 polymers-15-02349-f006:**
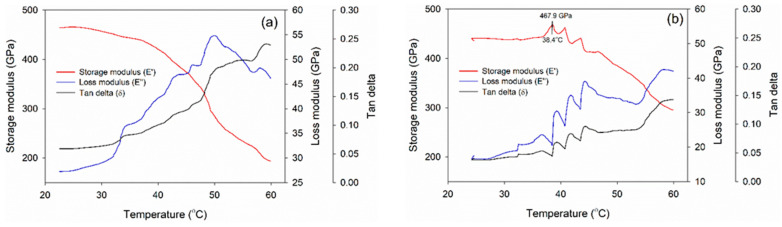
DMA curve of the cold-setting melamine-based adhesives: (**a**) control (MF-0) and (**b**) modified (MF-1).

**Figure 7 polymers-15-02349-f007:**
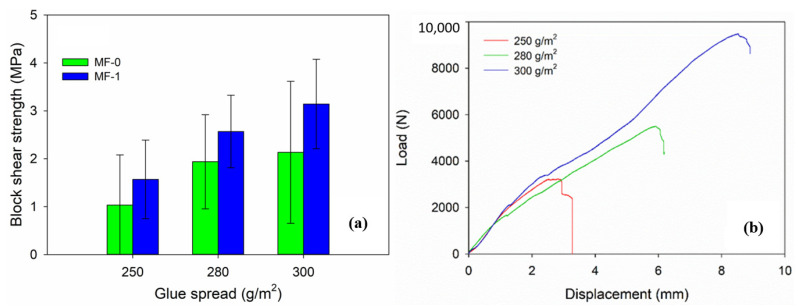
Block-shear strength of CLT bonded with melamine-based adhesive at different glue spreads in the control (MF-0) and modified MF (MF-1) (**a**) and the typical load versus displacement curve of the block-shear strength of MF-1 (**b**).

**Figure 8 polymers-15-02349-f008:**
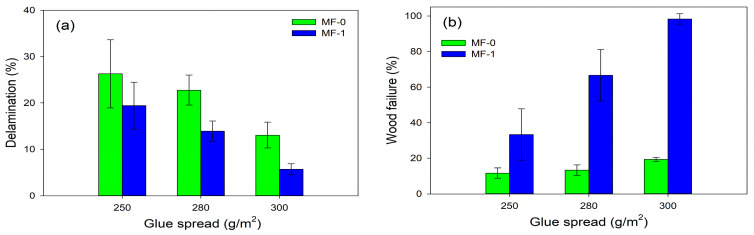
Delamination (**a**) and wood failure (**b**) percentage of CLT bonded with MF-0 and MF-1 at different glue spreads.

**Figure 9 polymers-15-02349-f009:**
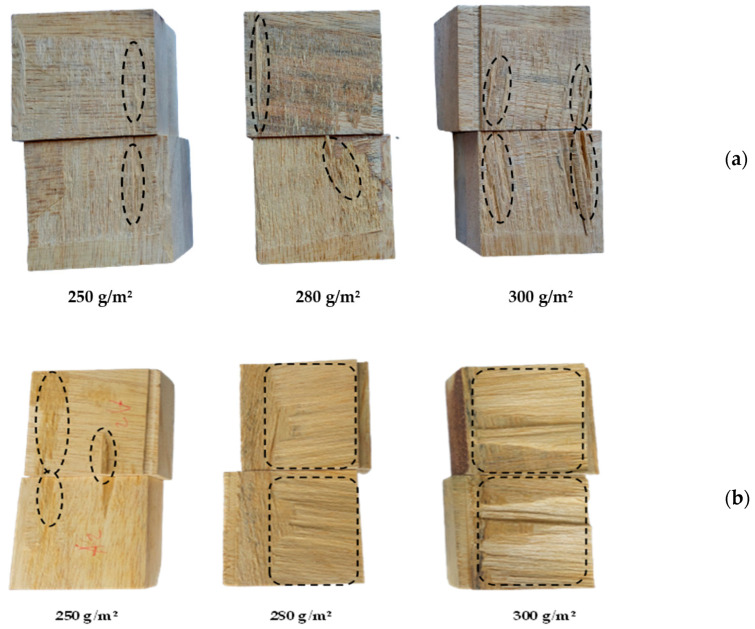
Images of wood failure after the block-shear test of CLT bonded with MF-0 (**a**) and MF-1 (**b**) at different glue spreads.

**Table 1 polymers-15-02349-t001:** Basic properties of cold-setting melamine-based adhesives.

Type of Adhesive	Properties			
	Solids Content (%)	Gelation Time (min, T = 100 °C)	Viscosity (mPa∙s, T = 25 °C)	pH
MF-0	51.89 ± 0.36	30.27 ± 0.15	484.51 ± 3.69	7.67 ± 0.58
MF-1	48.56 ± 0.08	6.90 ± 0.10	570.75 ± 14.80	6.67 ± 0.58

**Table 2 polymers-15-02349-t002:** Analysis of variance (ANOVA) and Duncan’s multiple range test of CLT with MF-1 adhesive properties at different glue spreads.

Properties	SS	Df	MS	F-Value	*p*-Value	R^2^
Bonding strength	1796.7540	1	1796.7540	6.2785	0.0406	0.4728
Delamination	2944.8490	1	2944.8490	24.1056	0.0017	0.7749
Wood failure	3317.1950	1	3317.1950	48.0947	0.0002	0.8729

## Data Availability

The authors confirm that the data underlying the research are included in the article. The raw data that support the results are available upon reasonable request from the corresponding author.

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
