# Peer review of "Effect of Glue Spread on Bonding Strength, Delamination, and Wood Failure of Jabon Wood-Based Cross-Laminated Timber Using Cold-Setting Melamine-Based Adhesive"

_polymers, 2023, doi:10.3390/polym15102349_

Round 1

Reviewer 1 Report

line 103- 5% and 3% pMDI were stated in the preparation of the adhesive, but only 3% was used for the tests. Why was there no data comparing them? 

line 194- this sentence is hard to understand because it is grammatically incorrect. Please correct this.

line 286- since the MF-0 could not be heat cured in the CLT, perhaps a better control would be to add a catalyst so a cured control could be compared to the MF-1.

line 350- the delamination should be "less than 10%", not at least 10%, and the wood failure should be at least 90%.

Author Response

Thank you for the constructive comments and suggestions. The manuscript has been revised according to the suggestions.  The appropriate corrections and modifications to the revised manuscript are presented with different highlighted fonts each time.

  • Line 103:  5% and 3% pMDI were stated in the preparation of the adhesive, but only 3% was used for the tests. Why was there no data comparing them? 

Thank you for your comment and question.

In our study, we did not compare the data of pMDI concentration because we only work with one pMDI concentration of 5% (based on the solid content of control adhesive, MF resin) as we describe in lines 102-104 of our manuscript. At the same time, 3% (based on the solid content of the control adhesive) is the concentration of citric acid used in our study.

We have checked and corrected the sentences in our manuscript, so the sentences are as follows:

In addition, based on the solids content of the control adhesive, the citric acid (20 wt%) of 5% as a catalyst and pMDI (96.2 wt%) of 3% as cross-linker were also added to the melamine-based adhesive.

  • Line 194: this sentence is hard to understand because it is grammatically incorrect. Please correct this.

Thank you for your advice, we have checked and corrected the sentences in our manuscript, so the sentences are as follows (line 202):

 “The addition of citric acid, pMDI, and wheat flour have lead the balance of the physical and chemical properties on melamine resin to reduce the solid content of the resin. The solid content of MF-1 decreased after adding citric acid, pMDI, and wheat flour to the MF-0.

  • Line 286: since the MF-0 could not be heat cured in the CLT, perhaps a better control would be to add a catalyst so a cured control could be compared to the MF-1.

Thank you for your comment and advice.

The MF-0 itself could not be used as a cold-setting adhesive therefore the CLT should be prepared using hot-pressing (heat treatment). We just completed manufacturing a control CLT bonded with MF-0 at different glue spreads of 250, 280, and 300 g/m2. The MF-0 CLT was manufactured using a cold-press under 1 MPa while clamped for 2-4 minutes and then subjected to heat treatment in the oven at 100 degrees for 2 h (Figure 1). The bonding performance of CLT bonded with MF-0 was compared to that of MF-1, as shown in Figure 7-9. 

  • Line 350: the delamination should be "less than 10%", not at least 10%, and the wood failure should be at least 90.

Thank you for your comment.

We have checked and corrected the sentences in our manuscript, so the sentences are as follows (line 352).

The standard specifies that total delamination and wood failure percentage should be less than 10%, and at least 90%.

Reviewer 2 Report

1. Poor hypothesis. Thermo-set adhesives are restricted by the poor thermal transmission of thick wood panels.

2. The effect of 5% citric acid or 3% pMDI on the properties of MF adhesive should be investigated.

3. How to prepare the samples for FTIR and DMA analysis?

4. A concise conclusion is necessary, there are too many results.

Author Response

Thank you for the constructive comments and suggestions. The manuscript has been revised according to the suggestions.  The appropriate corrections and modifications to the revised manuscript are presented with different highlighted fonts each time.

  • Poor hypothesis. Thermo-set adhesives are restricted by the poor thermal transmission of thick wood panels.

Thank you for your comment.

Yes, that's right, I agree with your opinion. The thermal transmission is dependent on thickness, as well as the thermal resistance properties of the material.  The thicker material generally has a higher thermal resistance and low (poor) thermal transmission.

The low thermal transmission value of the material is an advantage for its use as a thermal insulation material. High thermal transmission values, on the other hand, are desirable for rapid heat transfer, such as curing adhesives in constructing items comprised of wood layers. So far, MF-based adhesives have been widely used in the manufacture of plywood and with hot pressing applications.  Thin veneer layers as a plywood constituent material cause good thermal transmission from the hot-press, thereby helping to speed up the curing process of adhesive in the manufacture of plywood.

 Meanwhile, Pizzi (2003) reported that MF-based adhesive can be used as a cold-setting wood laminating adhesive for glulam and finger jointing by the use of adequate acid hardeners. So in this work, we tried to modify MF adhesive by mixing with acetic acid as a catalyst and pMDI as a cross-linker of cold-setting MF-based adhesive for CLT manufacturing. We want to study the characteristics and qualities of MF-based adhesives for cold pressing applications and to develop these adhesives for CLT production.

  • The effect of 5% citric acid or 3% pMDI on the properties of MF adhesive should be investigated.

Thank you for your advice.

In this study we prepared commercial MF adhesive (MF-0) as a control; modified MF (MF-1) with an addition of 5% citric acid and 3% pMDI as a cold-setting MF-based adhesive.

We have investigated the effect of 5% citric acid or 3% pMDI on the properties of MF as described in the manuscript (lines 202-243).

The addition of 5% citric acid and 3% pMDI increased the viscosity and cohesion strength of the melamine-based adhesive while decreasing the solids content, gelation time, and pH.

The viscosity of MF-1 was higher than MF-0 due to the formation of a bigger molecular chain from the interaction of MF, citric acid, and pMDI. The cohesion strength is directly proportional to the adhesive, while the viscosity adhesive is related to cohesion strength. The cohesion strength of MF-1 was higher than that of MF-0. CLT bonded with MF-1 is expected to have better properties than MF-0.

The gelation time of MF-0 was five times longer than MF-1. The MF-0 did not contain a citric acid catalyst or pMDI cross-linker, while MF-1 was prepared by incorporating MF-0, citric acid, and pMDI. The addition of a catalyst and a cross-linker speeds up the gelation time process by creating a larger molecular chain.

  • How to prepare the samples for FTIR and DMA analysis?

We have described the procedure for FTIR and DMA analysis in the manuscript, part 2.3 (lines 132-145).

Prepared sample for FTIR analysis:

The MF-0 and MF-1 specimens were scanned at 400-4000 cm-1 wavenumbers, 4 cm11 resolution, and 16 scans per sample in the sample holder. To normalize the adhesive spectrum, a min-max normalization was performed (Ver. 10.5.3, PerkinElmer Inc.).

Prepared sample for DMA analysis:

Each adhesive was used to bond two Whatman filter papers with a glue spread of 300 g/m2 to create a specimen measuring 50 mm 8 mm 0.2 mm. Before the DMA analysis, all specimens were precured in an oven at 50°C for 5 minutes.

The storage modulus (E), loss modulus (E′′), and tan delta min-max specimen were determined in the dual cantilever mode at a frequency of 1 Hz, strain level of 0.01%, and heating rate of 5°C/min in the scanning range of 20-60 °C.

  • A concise conclusion is necessary, there are too many results.

Thank you for your advice.

We have revised the conclusion in our manuscript as follows (lines 385-394).

This research investigated the influence of 250, 280, and 300 g/m2 glue spread on bonding strength, delamination, and wood failure percentage of jabon CLT. The results showed that glue spread had a greater impact on wood failure than bonding strength and delamination with an R2 of 0.8729. According to the multiple range tests of Duncan, a glue spread of 300 g/m2 could produce CLT with better bonding strength, delamination, and wood failure than 250 g/m2 and 280 g/m2.

At 300 g/m2, MF mixed with 5% and 3% citric acid and pMDI could be used as a cold-setting adhesive in CLT manufacturing. The addition of 5% citric acid and 3% pMDI increased the viscosity and cohesion strength of the melamine-based adhesive while decreasing the solids content, gelation time, and pH.

Reviewer 3 Report

Interesting article about the bonding of exotic Jabon wood. Popular topic but, due to the choice of exotic wood, it is especially important where it is popular.
Correct research described, although these descriptions could be extended a bit. Valuable results were obtained, consistent with the literature. The descriptions of the analysis of the results are sometimes hard to understand, especially when the sentences describing them are long. I recommend using shorter, simpler sentences.

The terminology for the description of the resin setting process and the formation of large molecules should be corrected. I suggest analyzing the terms molecular chain and polymer.

Is it really "The standard 349 specifies that total delamination and wood failure per-centage should be at least 10% and 90%, respectively" - please think about it.

There are a few minor editing errors in the work, some of them have been marked in the text of the manuscript.
The English language requires minor grammar corrections.

Author Response

Thank you for the constructive comments and suggestions. The manuscript has been revised according to the suggestions.  The appropriate corrections and modifications to the revised manuscript are presented with different highlighted fonts each time.

  • Interesting article about the bonding of exotic Jabon wood. Popular topic but, due to the choice of exotic wood, it is especially important where it is popular.
    Correct research described, although these descriptions could be extended a bit. Valuable results were obtained, consistent with the literature. The descriptions of the analysis of the results are sometimes hard to understand, especially when the sentences describing them are long. I recommend using shorter, simpler sentences.

Thank you for your comment and advice.

We have revised some sentences that were too long in order to make them shorter, simpler, and easier to understand.

  • The terminology for the description of the resin setting process and the formation of large molecules should be corrected. I suggest analyzing the terms molecular chain and polymer.

Thank you for your suggestion. We have revised the manuscript accordingly.

  • Is it really "The standard (line 349) specifies that total delamination and wood failure percentage should be at least 10% and 90%, respectively" - please think about it.

Thank you for your comment.

We apologize, there is a typo in the sentence, so the meaning is unclear. The sentence has been corrected as follows (line 352):

The standard specifies that total delamination and wood failure percentage should be less than 10%, and at least 90%.”

  • There are a few minor editing errors in the work, some of them have been marked in the text of the manuscript. The English language requires minor grammar corrections.

Thank you for your comments.

We appreciate all of your feedback and suggestions. We have revised the manuscript in accordance with your suggestions.

Round 2

Reviewer 3 Report

Dear Authors,

I'm glad my comments were taken into account.

After making changes, the article makes a much better impression. In my opinion, Conclusions still needs a minor correction. The authors refer to specific results. I think generalizations should be added.

Author Response

Dear reviewer,

Thank you for your comment and valuable advice.

We have revised the conclusion as per your suggestion to make it more general. Please find as followed the statement of the revised conclusion.  

"This research developed the CLT from jabon wood made with the modified MF adhesive by adding 5% citric acid and 3% pMDI for cold-setting application. This addition has increased the viscosity and cohesion strength of the melamine-based adhesive and decreased the solids content, gelation time, and pH. The CLT with various glue spreads of 250, 280, and 300 g/m2 were investigated to characterize the bonding strength, delamination, and wood failure percentage. The findings indicated that a higher glue spread led to increased bonding strength, lower delamination values, and higher wood failure percentages as indicators of the high performance of the bondability system. The statistical analysis found that the glue spread has a more significant impact on wood failure than bonding strength and delamination. Through our study, the MF adhesive mixed with citric acid and pMDI can be a promising cold-setting adhesive for future CLT production which considers the heat energy consumption.
